# Drug-Induced Resistance and Phenotypic Switch in Triple-Negative Breast Cancer Can Be Controlled via Resolution and Targeting of Individualized Signaling Signatures

**DOI:** 10.3390/cancers13195009

**Published:** 2021-10-06

**Authors:** Swetha Vasudevan, Ibukun A. Adejumobi, Heba Alkhatib, Sangita Roy Chowdhury, Shira Stefansky, Ariel M. Rubinstein, Nataly Kravchenko-Balasha

**Affiliations:** The Institute of Biomedical and Oral Research, The Hebrew University of Jerusalem, Jerusalem 9112102, Israel; swetha.vasudevan@mail.huji.ac.il (S.V.); ibukun.adejumobi@mail.huji.ac.il (I.A.A.); heba.alkhatib@mail.huji.ac.il (H.A.); sangita.roychowdhu@mail.huji.ac.il (S.R.C.); shira.stefansky@mail.huji.ac.il (S.S.); ariel.rubinstein@mail.huji.ac.il (A.M.R.)

**Keywords:** triple negative breast cancer, patient-specific altered signaling signatures, precision medicine, anti-EGFR therapy, targeted therapy, drug resistance

## Abstract

**Simple Summary:**

Patients with Triple Negative Breast Cancer (TNBC) have a poor prognosis due to high inter-tumor heterogeneity and absence of effective targeted treatments. Through quantification of ongoing processes in each individual with TNBC, we propose an explanation on why certain previously suggested monotherapies, such as anti-EGFR, are not effective. We experimentally demonstrate that monotherapies or drug combinations that are not adjusted accurately to the patient-specific ongoing processes may create an evolutionary pressure on a tumor leading to the emergence of previously undetected or untargeted cellular subpopulations. We show for example that certain TNBC tumors may benefit from therapies targeting estrogen receptors (ER), similarly to ER positive cancers. When untargeted, those tumors may develop large ER positive subpopulations. We propose that anti-TNBC therapy should be accurately tailored to the personalized molecular processes and that incomplete or “wrong” treatments may generate diverse evolutionary routes of TNBC tumors leading to drug resistance.

**Abstract:**

Triple-negative breast cancer (TNBC) is an aggressive subgroup of breast cancers which is treated mainly with chemotherapy and radiotherapy. Epidermal growth factor receptor (EGFR) was considered to be frequently expressed in TNBC, and therefore was suggested as a therapeutic target. However, clinical trials of EGFR inhibitors have failed. In this study, we examine the relationship between the patient-specific TNBC network structures and possible mechanisms of resistance to anti-EGFR therapy. Using an information-theoretical analysis of 747 breast tumors from the TCGA dataset, we resolved individualized protein network structures, namely patient-specific signaling signatures (PaSSS) for each tumor. Each PaSSS was characterized by a set of 1–4 altered protein–protein subnetworks. Thirty-one percent of TNBC PaSSSs were found to harbor EGFR as a part of the network and were predicted to benefit from anti-EGFR therapy as long as it is combined with anti-estrogen receptor (ER) therapy. Using a series of single-cell experiments, followed by in vivo support, we show that drug combinations which are not tailored accurately to each PaSSS may generate evolutionary pressure in malignancies leading to an expansion of the previously undetected or untargeted subpopulations, such as ER+ populations. This corresponds to the PaSSS-based predictions suggesting to incorporate anti-ER drugs in certain anti-TNBC treatments. These findings highlight the need to tailor anti-TNBC targeted therapy to each PaSSS to prevent diverse evolutions of TNBC tumors and drug resistance development.

## 1. Introduction

Triple-negative breast cancer (TNBC) is an aggressive subgroup of breast cancers which accounts for 10–20% of all breast cancers diagnosed. These tumors are characterized by low expression of Estrogen (ER), Progesterone (PR) and Her2 receptors [1]. TNBC has a poor clinical outcome and higher proliferative rate when compared to other breast cancer subtypes [2]. At present, there is no optimal protocol for the treatment of triple-negative breast cancer [3]. This is due to their aggressive phenotype and lack of the molecular druggable biomarkers [3]. Chemotherapy, such as Taxol, is often used to treat TNBC patients, who often develop complications due to the toxic effects of the drug. Therefore, there is an unmet need to develop targeted therapies for the treatment of TNBC.

Tremendous efforts are being invested in identifying potential oncogenic biomarkers, both at the genetic and proteomic levels, with the aim to identify drug targets [4,5,6]. Epidermal growth factor receptor (EGFR), a transmembrane receptor belonging to the EGFR family, is detected frequently and at high levels in TNBC cancers [7] and thus represent a useful therapeutic target for TNBC [8,9]. EGFR plays an important role in promoting uncontrolled cell proliferation and opposing apoptosis [10]. EGFR amplification is further associated with poor prognosis, resistance to both chemotherapy and radiotherapy, increased risk of recurrence and metastasis, and reduced overall and disease-free survival [11]. Nevertheless, clinical trials of EGFR inhibitors have failed due to the rapid development of resistance.

Several theories were suggested to explain the failure of anti-EGFR therapies. One of the theories stated that the role and dominance of EGFR signaling may change during tumor progression [12]. Additionally several clinical studies demonstrated that metastatic TNBC rarely depends on EGFR signaling alone for survival [13,14,15,16]. Other signaling pathways, such as MAPK and AKT, may act simultaneously in tumors and compensate for the loss of EGFR signaling [17].

In this study, we aimed to characterize EGFR expression in the context of patient-specific protein–protein co-expression networks in order to suggest a possible mechanism for the poor performance of anti-EGFR therapies in TNBC. We utilized an information-theoretic surprisal analysis (SA) to resolve a patient-specific signaling signature (a PaSSS) for each tumor as described previously [18]. The analysis revealed that 747 breast tumors could be characterized by 17 altered protein–protein subnetworks, named unbalanced processes. Each PaSSS was characterized by a patient-specific subset of 1–4 processes out of 17. We suggest that a key for promising therapies lies in an accurate resolution of the PaSSS in each TNBC. All patient-specific processes should be inhibited, through targeting one or two central protein hubs in each process, in order to reduce the tumor-specific biochemical flux [19]. PaSSS analysis suggested that only a small part of TNBC tumors (~30%) would benefit from anti-EGFR therapies and only when they are combined with anti-ER therapies. This combination may include an additional 3rd drug when a PaSSS consists of more than two processes. In contrast, anti-ER therapies, mono/or as a part of combined therapy, were predicted to be efficient in more than 80% of TNBC patients.

Experimental validation, using TNBC cell lines harboring EGFR+ and EGFR− PaSSSs, respectively, have shown that the PaSSS-based therapies were highly efficient and selective in preventing drug resistance development. The PaSSS-based combination for EGFR+ MDA-MB-468 TNBC cell line, comprising of anti-EGFR, anti-ER and anti-MAPK inhibitors stopped the MDA-MB-468 tumor growth, but not the growth of another TNBC, MDA-MB-231 cells, the PaSSS of which did not include EGFR. The growth of MDA-MB-231 malignancy was inhibited by a combination, comprised of anti-MAPK and 2-deoxyglucose (2DG) inhibitors, as predicted by PaSSS analysis.

Furthermore, single-cell quantification of distinct subpopulations, independently evolving in TNBC malignancies in response to different treatments, revealed that monotherapies, such as anti-EGFR or combined therapies, which were predicted to target only partially the identified PaSSSs, induced expansion of cellular subpopulations harboring either untargeted protein subnetworks or initially inactive pathways, such as ER+ or AKT+. Simultaneous targeting of all unbalanced processes within the PaSSS was crucial to prevent phenotypic change, such as a switch from TNBC to ER+ tumor phenotype, or a switch from one signaling state to another that enables sustaining of tumor growth.

## 2. Materials and Methods

### 2.1. Cell Lines and Culture

MDA-MB-468 (TNBC) and MDA-MB-231 (TNBC) were obtained from the American Type Culture Collection (ATCC, Manassas, VA, USA) and authenticated by the Genomic Center of the Technion Institute (Haifa, Israel). Both cell lines were grown in RPMI-1640 medium with 10% fetal bovine serum (FBS), supplemented with 4 mM L-glutamine and Pen-Strep (100 U/mL Penicillin and 100 μg/mL Streptomycin) in a 37 °C incubator (5% CO_2_). All media and supplements were from Biological Industries, Israel. Cells were checked for the absence of mycoplasma contamination.

### 2.2. Survival Assay

Cells were seeded and allowed to grow to about 60% confluency and were subsequently treated with various drug combinations predicted by PaSSS analysis for 72 h. The medium was changed twice a week while observing the cells for regrowth or repopulation at various time points (7 days, 14 days, 21 days, etc.). The cells were washed once with PBS and fixed with 4% paraformaldehyde at room temperature for 30 min. They were then, stained with methylene blue, washed, and left overnight to dry. Color extraction was done by adding 0.1 M hydrochloric acid for 1 h at room temperature. The absorbance was read 630 nm. The results were normalized by the day zero, untreated control.

### 2.3. Cell Fixation and Permeabilization of Phosphoproteins for Flow Cytometry

5 × 10^5^ cells were seeded in a medium flask and were treated accordingly with the various therapies for 72 h and then allowed to regrow. The cells were collected at various time points by enzymatic detachment with accutase. They were fixed with pre-warmed 2% PFA at 37 °C for 10 min [20]. The cells were then washed by adding PBS to the cell suspension and centrifuged at 300 rcf for 3 min. The supernatant was discarded and the pellets were re-suspended in a permeabilization buffer (BD Phosflow™ Perm Buffer III, Becton, Dickinson and Company, Franklin Lakes, NJ, USA; cat. No. 558050) for 30 min on ice. The cells were washed again as previously described. The pellets were then re-suspended in FACS buffer (2% BSA in PBS) and stored in −80 °C.

### 2.4. Flow Cytometry Analysis

Cells were labeled with fluorescently tagged antibodies. The following conjugated antibodies were used: anti-p-EGFR (Y1068) (R&D Systems, Minneapolis, MN, USA, cat. no. IC3570G), anti-p-ERK2 (Thr202/Tyr204) (BioLegend, San Diego, CA, USA, cat. No. 675503), anti-p-S6 (Ser235/236) (BioLegend, cat. no. 608605), and anti-GAPDH (Santa Cruz Biotechnology, Dallas, Texas, USA, cat. no. sc-47724AF594). Anti-p-AKT (Ser473) (Rockland, Limerick, PA, USA, cat. no. 200-301-268) was conjugated to PerCP/Cy5.5 Conjugation kit (Abcam, Cambridge, MA, USA, ab102911) and anti-p-Estrogen alpha Ser118 (Biorbyt, San Francisco, CA, cat. no. orb6021) to DyLight™ 405 (Jackson ImmunoResearch, West Grove, PA, USA, 711-475-152. Compensation control was done using UltraComp eBeads (Thermo Fisher Scientific, Waltham, MA, USA,). Single-cell suspensions were analyzed using LSR-Fortessa Analyzer. The number of events (cells) profiled for each sample was ranging from 30,000–50,000 cells. Preliminary data analysis was done using FlowJo software (Version 10, Becton, Dickinson and Company, Ashland, OR, USA) and the output data were extracted into an Excel file.

### 2.5. Animal Studies

MDA-MB-468 (1 × 10^6^ cells/mouse) or MDA-MB-231 (1 × 10^6^ cells/ mouse) were inoculated orthotopically into 7-week old female NOD-SCID mice (at least *n* = 8 mice per group), and once the volume of the tumors reached 80 mm^3^, treatments were initiated 5 times a week for up to 7 weeks. Tumor volume was measured twice a week. Trametinib (Tr) (0.5 mg/kg) and Erlotinib (Er) (12.5 mg/kg) were suspended in aqueous mixture of 0.5% hydroxypropyl methylcellulose + 0.2% Tween 80 and administered by oral gavage. 2-Deoxy-glucose (2DG) (500 mg/kg) and Taxol (20 mg/kg) were suspended in saline and injected intraperitoneally. Taxol was administered once a week. Tamoxifen (Tam) (10 mg/kg) was suspended in corn oil and was also injected intraperitoneally 3 times a week. All the drugs were purchased from Cayman Chemicals (Ann Arbor, MI, USA). The Hebrew University is an AAALAC International accredited institution. All experiments were conducted with approval from the Hebrew University Animal Care and Use Committee.

### 2.6. Western Blot Analysis

Cells were seeded, treated and allowed to grow as mentioned above in the survival assay. The samples were lysed at different time points and Western blot analysis was performed as described in [21]. The following antibodies were used—anti-p-S6 (Ser235/236) (cat. No. 4858S; 1:1000), anti-p-AKT (Ser473) (cat. No. 4060S; 1:1000), anti-p-IGFR (Y1131) (cat. No. 3021; 1:1000), and anti-p-EGFR (Y1068) (cat. No. 3777S; 1:1000). The antibodies were purchased from Cell Signaling Technology, Inc. (Danvers, Massachusetts, USA). Anti-p-ERK2 (E4) (cat. No. SC7383; 1:200) and anti-total-GAPDH (cat. No. SC47724; 1:200) antibodies were purchased from Santa Cruz Biotechnology. Original Western blots are presented in Appendix A.

### 2.7. PaSSS Analysis

The TCGA dataset comprising of 747 breast tumors was a part of a large dataset consisting of 3467 human tumors, which were profiled on a reverse-phase protein array (RPPA) for 181 cancer associated proteins, and analyzed using surprisal analysis as described in [18,19,22]. The input data was obtained from TCPA (The Cancer Proteome Atlas) portal.

Briefly, surprisal analysis, which is an information-theoretical approach, uses thermodynamic laws [23] to study altered cellular networks in a patient-specific manner. The approach identifies unbalanced processes in a biological system, which deviate the system from its balanced, steady-state. Every tumor sample is considered a complex biological system for which a set of altered molecular processes or *unbalanced processes* is identified.

Using protein expression levels, the analysis quantifies the expected expression levels for every protein *i*, at the steady and deviations thereof using the equation:(1)$$lnX_{i}\left(k\right)=lnX_{i}^{o}\left(k\right)-\underset{\alpha =n}{\sum }G_{i\alpha }\lambda_{\alpha }\left(k\right)$$
where $lnX_{i}^{o}\left(k\right)$ is the logarithm of the expression level of protein *i* at the balanced state, and the sum, $\sum_{\alpha =n}G_{i\alpha }\lambda_{\alpha }\left(k\right)$, represents the deviations from the balanced state level [23,24,25]. The term *G_iα_* denotes the weight or extent of the participation of each individual protein *i* in the specific unbalanced process *α*, while *λ_α_(k)* indicates the amplitude (importance) of the unbalanced process, in every tumor *k.* Their sign indicates the correlation or anti-correlation between proteins in the same process and between different tumors in the same process, respectively. To find the actual change in expression level for each protein *i* in the tumor *k*, the contribution of each ongoing process *G_iα_λ_α_(k)* is calculated, which can be positive or negative. It is important to note that each protein can participate in many unbalanced processes. In this study, we analyze changes in the proteomic processes as described previously [18,19,22].

Not all tumors are influenced by all unbalanced processes. So for each tumor, a set of unbalanced processes is generated, which is converted into a patient-specific barcode as described in [18,19,22]. Briefly, for each patient, *λ_α_(k)* (*α* = 1, 2, 3, …, *n*) values are normalized as follows: If *λ_α_(k)* > 2 (and is therefore significant according to calculation of threshold values [22]) then it was normalized to 1; if *λ_α_(k)* < −2 (significant according to threshold values as well) then it was normalized to −1; and if −2 < *λ_α_(k)* < 2 then it was normalized to 0. Each patient is assigned then a barcode using −1, 1 and 0 values—which represents a patient-specific combination of the active processes (Appendix A). This combination is a patient-specific signaling signature that can be used further to devise patient-specific drug therapies [18,19] as shown in Figure 2. For more details see Appendix A describing the flow of the analysis and references [18,19].

### 2.8. Selection of TNBC Cell Lines for Experimental Validation of the PaSSS-Based Strategy

In this study we selected 2 TNBC cell lines MDA-MB-231 and MDA-MB-468, which were part of the dataset analyzed by us previously [18] to validate the approach. To provide a statistical meaning for the selection of 2 TNBC cell we calculated an upper bound for the probability to select 3 and 5 unbalanced processes found in MDA-MB-231 and MDA-MB-468 cells, randomly. Calculation of the upper bound for the probability to select randomly 3 or 5 unbalanced processes was based on the frequency of the most abundant unbalanced processes in the breast cancer subset (Appendix A of the reference [18]). The probability to find the three most abundant processes in a particular breast cancer sample equals to:(*152*/747) × (*179*/747) × (*563*/747) = 0.036749.(2)

Numbers in italic represent numbers of breast cancer patients found to harbor the most abundant processes in the subset (Appendix A and [18]), e.g., processes 1, 2, and 3 and the number 747 is the number of breast cancer patients in this subset. The probability to find the five most abundant processes in a particular breast cancer sample was calculated in a similar manner and equals to 0.000149. Thus the upper bound for the probability to select 3 and 5 unbalanced processes, characterizing MDA-MB-231 and MDA-MB-468 cells, randomly equals 5.47442 × 10^−6^, which is <<0.001.

### 2.9. Computational Single Data Analysis for Calculation of Cell-Specific Signaling Signatures (CSSS)

Single-cell analysis is an adaptation of bulk surprisal analysis to single cells [26]. The analysis allows the partitioning of a tumor mass into distinct, independently evolving cellular subpopulations and characterization of the altered protein networks (unbalanced processes) associated with each subpopulation [27]. The analysis allows identifying of very small, previously undetected subpopulations, which are hardly detected in bulk assays, such as cancer stem cells and/or drug-resistant cells.

TNBC cells were treated with various drug combinations as described in Figures 3–5 and were allowed to regrow (Appendix A). The regrown cellular populations were labeled with fluorescently tagged antibodies against the central proteins representing the unbalanced processes identified in the bulk analysis (pEGFR, pERK, pER, pS6, pAKT, and GAPDH) (Appendix A and [18]). The simultaneous labeling of cells with these antibodies enabled us to examine if those processes were expressed together in the same cells or represented different subpopulations [26,27]. For each experimental condition we profiled 30,000–50,000 single cells.

The input matrix for the single-cell analysis includes the expression levels of proteins quantified in single cells. For each protein, the number of the processes influencing its level is computed (Appendix A) using the following equation [24]:(3)$$\underset{\begin{array}{c} \mathrm{experimental}\\ \mathrm{level}\mathrm{of}\mathrm{protein}i \end{array}}{\underset{︸}{X_{i}\left(cell,t\right)}}=\underset{\begin{array}{c} \mathrm{level}\mathrm{of}\mathrm{protein}i\\ \mathrm{in}\mathrm{the}\mathrm{reference}\mathrm{state} \end{array}}{\underset{︸}{X_{i}^{o}\left(cell,t\right)}}exp\underset{\begin{array}{c} \mathrm{changes}\mathrm{in}\mathrm{protein}\mathrm{levels}\\ \mathrm{due}\mathrm{to}\mathrm{the}\mathrm{constraints}\alpha =1,2,\ldots \end{array}}{\underset{︸}{\left(-\sum_{\alpha =1}G_{i\alpha }\lambda_{\alpha }\left(cell,t\right)\right)}}.$$

Proteins that deviate from the steady state in the same direction are grouped into subnetworks (unbalanced processes). As a next step, the analysis assigns a cell-specific *set* of processes named cell-specific signaling signature (CSSS) for each cell. A detailed description of the procedure is described in reference [26]. Hence, CSSS divides the tumor mass into intratumor subpopulations which are defined as a group of cells harboring the same set of unbalanced processes, or the same CSSS. Each CSSS is schematically converted into a barcode (Appendix A) where white squares mean inactive and black squares mean active processes in a cell. We then define a cellular subpopulation as a group of cells harboring the same CSSS [26] (Appendix A).

## 3. Results

### 3.1. Resolution of the Signaling Structures in Breast Cancer Patients Suggests That Anti-EGFR Monotherapy Should Be Inefficient in TNBC in Contrast to Anti-ER Therapies

Using a subset of 747 breast cancer tissues from the previously published TCGA dataset [28], which included 3467 tumors profiled for 181 functional oncoproteins each, we examined EGFR/pEGFR expression levels in TNBC and non-TNBC tumors. No significant difference in EGFR/pEGFR expression levels was found between TNBC and non-TNBC tumors (Figure 1A). Examination of the protein–protein co-expression patterns in breast cancer subset, which were determined using information-theoretic surprisal analysis (Appendix A, [18], Appendix A), revealed that EGFR protein could be found in several altered sub-networks that characterized both TNBC and non-TNBC tumors. Seventeen altered protein subnetworks (Methods, Appendix A, Appendix A), named unbalanced processes, repeated themselves in the breast cancer subset. Each breast, TNBC/non-TNBC, tumor was found to harbor a subset of 1–4 distinct processes out of 17, representing a patient-specific signaling signature, (a PaSSS). An example for three different PaSSSs, characterizing three TNBC tumors, is shown in Figure 1B. For the simplicity of representation, each PaSSS was schematically transformed into a barcode (Figure 1B, Appendix A). We found that the TNBC subgroup was relatively heterogeneous, as 74 samples were characterized by 17 different PaSSSs leading to a higher heterogeneity index (Figure 1C), in comparison with the less heterogeneous non-TNBC BRCA subset (673 samples) which were characterized by 84 PaSSSs (Figure 1C).

Central protein/s from each sub-network were selected in order to assign a combination of FDA-approved targeted drugs for each PaSSS ([18], Figure 1B, Appendix A, Appendix A). The PaSSS-based approach suggested that 23 out of 74 (31%) TNBC tumors (Appendix A) would benefit from anti-EGFR drugs only when they are combined with anti-ER therapies (e.g., tamoxifen) in all cases and in certain cases with an additional drug when PaSSSs comprise more than two processes (see examples in Figure 1B: patients TCGA-E9-A295 and TCGA-BH-A0DI and see Appendix A for the complete list). However, the majority of TNBC PaSSSs did not include EGFR as a part of the altered signaling signature (see for example TCGA-BH-A0BW patient in Figure 1B and Appendix A). This finding corresponds to the recent study in which EGFR was quantified in 150 TNBC patient-derived tissues using immunohistochemistry in [29]. A similar result was obtained for non-TNBC tumors, in which only 31% of the tumors were suggested to benefit from either mono anti-EGFR therapy or from a combination of anti-EGFR with additional drugs (Appendix A).

These results suggest that TNBC patients would be unlikely to benefit from anti-EGFR monotherapies, as anti-EGFR monotherapies were predicted to either inhibit only partially the altered flux of TNBC tissues or to be ineffective. Interestingly, PaSSS analysis suggests that ~80% of TNBC patients should benefit from anti-ER therapy, applied as monotherapy or as a part of combined targeted therapy (Appendix A).

### 3.2. The PaSSS-Based Strategy Suggests How Anti-EGFR/Anti-ER Therapies Should Be Incorporated in Order to Reduce Efficiently the Individualized Signaling Fluxes and Cell Regrowth

To validate the PaSSS-based strategy, we selected two TNBC cell lines—MDA-MB-468 and MDA-MB-231 from the dataset of 10 different cell lines (Methods), PaSSSs of which were computed previously [18]. These cell lines represent the patient-derived TNBC subset from the TCGA dataset described above: the PaSSS of MDA-MB-468 included EGFR and estrogen receptor, ER, as central targets, thereby corresponding to ~31% TNBC tissues in the TCGA breast cancer subset (Figure 2A,B). MEK/ERK was suggested as an additional central target required for the efficient treatment of MDA-MB-468 cells ([18] and Figure 2A,B). Thus, we predicted that Erlotinib (anti-EGFR), Trametinib (anti -MEK/ERK) and Tamoxifen (Anti-ER; Er + Tr + Tam) should efficiently target the PaSSS of MDA-MB-468 malignancy. MDA-MD-231 cell line did not include EGFR as a part of its PaSSS, similarly to 69% of the TNBC patient-derived tissues. The suggested therapy for MDA-MB-231 cells included Trametinib and the glycolysis inhibitor 2-deoxyglucose (2DG) (Tr + 2DG) (Figure 2C,D and [18]).

We hypothesized that simultaneous targeting of all unbalanced processes in each PaSSS is required in order to prevent drug resistance development and cell regrowth.

### 3.3. Rationally Designed PaSSS-Based Drug Combinations Prevented the Development of Drug Resistance

We suggest that since the PaSSS-derived drug combinations are designed to target central elements of the individualized signaling networks simultaneously, they should significantly inhibit the drug-resistance development. This is in contrast to monotherapies, such as anti-EGFR, which are suggested to target only certain parts of the cancer networks (Figure 3A). To test this hypothesis, we treated both malignancies with either monotherapies or various combinations, which inhibit the signaling flux either partially or completely. We then allowed the cells to repopulate the plate and measured a degree of regrowth at various time points (7, 14, and 21 days). We found that when MDA-MB-468 cells were treated with either Erlotinib (Er) monotherapy or in combination with Tr, the cells regrew (Figure 3B). However, no regrowth was observed when Tam was added to Er + Tr, as predicted by PaSSS analysis. The importance of ER inhibition in TNBC MDA-MB-468 cells, was highlighted further in Figure 3C, showing that adding Tam to the cells, treated for 2 weeks with Er + Tr double therapy, significantly reduced the survival of those cells. However sequential addition of Tam, after 2 weeks of the treatment, had a weaker effect on the death of MDA-MB-468 cells in comparison with simultaneous application of the PaSSS-based, triple drug therapy.

These findings correspond to PaSSS analysis of the TCGA dataset, suggesting that certain TNBC patients should benefit from combinations of anti-EGFR and anti-ER therapies with or without additional drugs.

When a combination of Tr + 2DG, predicted for MDA-MB-231, was used to treat MDA-MB-468 cells, the cells did not respond to the treatment and started to regrow 7 days after the treatment (Figure 3B).

Similarly in MDA-MB-231 cells, partial inhibition of the signaling flux with either Tr, 2DG or Er monotherapies resulted in cell regrowth (Figure 3D). However, the PaSSS-based drug combination of Tr + 2DG prevented MDA-MB-231 cell regrowth. We also demonstrate that the drug combination predicted for MDA-MB-468 was less effective for MDA-MB-231 (Figure 3D).

Western blot analysis confirmed these results by demonstrating that the PaSSS-based combinations depleted entirely the intracellular signaling in MDA-MB-468 (Figure 3E and Appendix A) starting from day 3 of the treatment. Combination without Tam, which included Tr + Er only, was effective at the beginning, however at day 7 evoked pAKT, and then pEGFR at day 21. Cells that developed resistance to Er monotherapy were characterized by induced pAKT and pS6 (Figure 3E and Appendix A). Drug combination, Tr + 2DG, predicted for MDA-MB-231 cells was highly ineffective in treating MDA-MB-468 malignancy from the very beginning—day 3 (Figure 3B), as represented by induced activation of pAKT and pS6 starting from day 3 until day 21 (Figure 3E and Appendix A).

However, Tr + 2DG combination was significantly more effective in MDA-MB-231 cells (Figure 3F and Appendix A). The efficacy gradually increased until the depletion of MDA-MB-231 signaling at day 21. This result corresponds to the poor survival of the cells at this time point. The PaSSS-based combination predicted for MDA-MB-468 cells was very effective at the beginning for MDA-MB-231 cells. The result corresponded to the low survival of MDA-MB-231 cells at days 3 and 7. However, despite the low survival of the MDA-MB-231 cells at the beginning, the combination predicted for MDA-MB-468 induced pAKT activation starting from day 3. The activation also remained significant at day 21 (Figure 3F and Appendix A), corresponding to the regrowth of MDA-MB-231 cells at day 21 (Figure 3D).

These results suggest that the PaSSS-based drug combinations are effective and selective: the predicted and effective drug combination for one TNBC is significantly less effective for another and vice versa.

### 3.4. Monotherapies and Wrong Drug Combinations Do Not Deplete the Signaling of the TNBC Cell Lines but Induce a Switch from One Signaling State to Another

To further validate the induction of ER and AKT and to quantify the relationships between those proteins in response to monotherapies or incomplete drug combinations (e.g., Tr + Er in MDA-MB-468) we performed single-cell surprisal analysis (Methods). The analysis allowed to map the activity of these proteins within the cellular subpopulations evolving in response to different treatments. Additional central targets pS6, pEGFR, GAPDH and pERK (Figure 2), were added to the analysis. Using multicolor flow cytometry, expression levels of these proteins were quantified simultaneously in at least ~30,000 single cells in each treatment condition. The single-cell data was used to quantify protein–protein co-expression patterns (unbalanced processes) induced in response to treatments within different TNBC cellular subpopulations. This was achieved through identification of cell-specific signaling signatures (CSSS, Methods and Appendix A). Cells sharing the same CSSS were defined as a subpopulation.

Using CSSS analysis, we mapped subpopulations within the MDA-MB-468 cell population that evolved in response to Er, Er + Tr and Tr + 2DG. We found that following partial inhibition of the cells, targeting EGFR and ERK (Er + Tr), but not ER, subpopulations C and D, each harboring ER+ process (Figure 4B), expanded significantly at day 21 in response to Er + Tr (Figure 4C and Appendix A). The treatment led to an overall increase in the percentage of ER+ cells from 0.3% to 8.2% (Figure 4D). This result suggests that incomplete inhibition, which does not target ER, may lead to a switch from TNBC to ER+ phenotype and confirms the hypothesis that TNBC malignancies can benefit from anti-ER therapies.

CSSS analysis revealed an expansion of another subpopulation of cells in response to Er + Tr, subpopulation B, harboring process pAKT+/pS6− (Figure 4A–C). This subpopulation was almost undetected before the treatment and expanded up to 1.6% at day 21 (Figure 4C and Appendix A). This result corresponds to the induced activation of pAKT in response to Er + Tr as demonstrated in Figure 3E.

Tr + 2DG treatment led to an expansion of another AKT+ subpopulation, subpopulation A (Figure 4A–C). An overall increase in the percentage of pAKT+ cells is presented in Figure 4E,F. Interestingly in all treatment conditions the dominant pAKT+ and pER+ subpopulations evolved independently (Figure 4C). These results suggest that leaving certain elements in TNBC PaSSSs untargeted may enrich previously small subpopulations.

CSSS analysis of the MDA-MB-231 population revealed that Tr induced an expansion of the previously undetected subpopulations A, C, D, E, and F at day 7 (Figure 5C). Subpopulation E, harboring pEGFR+/pERK−, expanded from 0.4% to 8.2%, leading to a switch from EGFR− to EGFR+ phenotype (Figure 5E). Subpopulations A, B, C and D harboring pAKT+ processes (Figure 5A–C) increased significantly following either Tr monotherapy or Er + Tr + Tam combination treatment, leading to an overall increase in the percentage of AKT+ cells from 0.17% to >8% in Tr treated cells at day 7 and to >6% in Er + Tr + Tam treated cells at day 21 (Figure 5D).

Collectively, these results suggest that when the signaling flux is partially inhibited, it may lead to the clonal expansion of the previously undetected subpopulations of cells (as AKT+ subpopulations) or subpopulations harboring untargeted central proteins (such as ER+ subpopulations in MDA-MB-468 cells), leading to a switch from one signaling state to another.

### 3.5. The Predicted Drug Combinations Are Selective and Efficient in Preventing the Development of Tumor Regrowth In Vivo

We suggested that the PaSSS-based treatments should inhibit central elements of the individualized flux within the tumor, thereby significantly reducing the chance for the development of drug resistance. To validate the efficacy of the PaSSS-based drug combinations in vivo, MDA-MB-468 and MDA-MB-231 cells were orthotopically injected into the mammary fat pads of NOD SCID mice and treated five times a week for up to 7 weeks. Er, Tr, or Tam monotherapies could not inhibit the growth of MDA-MB-468 tumors (Figure 6A). MDA-MB-468 tumors responded initially to Er + Tr treatment, however, they relapsed 6 weeks later. The predicted drug combination of Er + Tr + Tam was very efficient and significantly inhibited the growth of MDA-MB-468 tumors (Figure 6A). Although Taxol sensitive, MDA-MB-468 tumors responded well to the chemotherapy, the PaSSS-based combination did not demonstrate poorer performance (Figure 6B). This suggests that the PaSSS-predicted combination was at least as effective as Taxol in the Taxol-sensitive MDA-MB-468 tumors. The predicted combination for MDA-MB-231 tumors was highly ineffective for MDA-MB-468 tumor (Figure 6A).

However, in the case of MDA-MB-231 tumors, Tr + 2DG showed an effect superior to all other therapies as predicted by PaSSS analysis (Figure 6C), including to Taxol, which was highly ineffective. These results demonstrated that the PaSSS-based combinations inhibit tumor growth in a patient-specific manner.

## 4. Discussion

Triple-negative breast cancer is an aggressive subtype of breast cancer for which an effective targeted therapy was not approved due to the lack of known druggable targets, such as estrogen, progesterone, and HER2 receptors [30]. Thus, there is an unmet need to develop new therapeutic strategies for TNBC. EGFR was considered to be frequently overexpressed in TNBC and to drive the disease progression [31]. However, efforts to target EGFR were associated with poor outcome in TNBC due to tumor heterogeneity and activation of alternative signaling pathways [9,32].

SA analysis of 747 breast tumors from the TCGA dataset suggested that ~30% of TNBC tumors should benefit from anti-EGFR inhibitors. Moreover, these inhibitors should be combined with anti-ER therapies. To validate this hypothesis, we selected two TNBC cell lines, MDA-MB-468 and MDA-MB-231 [18] that represented well two TNBC patient subpopulations in the TCGA dataset: first (31%), which was suggested to benefit from anti-EGFR/anti-ER inhibitors with or without additional drugs and the second subpopulation (69%), which does not harbor EGFR as a part of their PaSSSs. The second group was predicted to benefit from other targeted drug combinations. We suggested that the central proteins from all processes, comprising the PaSSS in each malignancy, must be targeted to collapse the entire signaling signature. We demonstrated in vitro and in vivo that the predicted drug combination of Er + Tr + Tam for MDA-MB-468 malignancy was superior to monotherapies or other drug combinations that were not predicted to target the PaSSS of MDA-MB-468. Similarly, in MDA-MB-231, the drug combination of Tr + 2DG collapsed efficiently the PaSSS it harbors and brought about the highest killing rate. In this TNBC anti-EGFR treatments were ineffective when applied as monotherapies or combined with the drugs predicted for MDA-MB-468.

Using a series of cell-regrowth, in vitro experiments followed by in vivo validation, we demonstrated that the rationally designed drug combinations were efficient and selective in inhibiting the tumor regrowth. A study by Hrustanovic et al. confirms further the importance of targeting multiple signaling pathways to overcome the development of resistance in cancer [33]. We take a step forward here by showing that the targeting of multiple signaling pathways is especially effective when is based on individualized PaSSSs. This corresponds also to our recent study demonstrating that the individualized therapy, tailored specifically to the personalized signaling signatures, could inhibit effectively and in a patient-specific manner the tumor growth in BRAF^V60E^ melanoma [19].

Single-cell CSSS analysis, which followed the temporal evolution of different subpopulations in response to incomplete or “wrong” drug combinations, validated our findings. We detected significant expansion of MDA-MB-468 subpopulations harboring activated ER in response to incomplete inhibition of the unbalanced flux with Er + Tr. This result corresponds to PaSSS analysis of the TCGA dataset, suggesting that certain EGFR+ TNBC tumors should benefit from the combined anti-EGFR and anti-ER inhibition. Additionally, we found that incomplete inhibition may activate anti-apoptotic pathways, such as the AKT pathway [34,35]. These findings correspond to the suggested mechanism of an adaptive response that might evolve in response to anti-EGFR therapy [36].

Similarly, we demonstrated that following incomplete inhibition of the signaling flux in MDA-MB-231 cells using either Tr monotherapy or the drug combination predicted for MDA-MB-468, the activity of the newly emerged pAKT+ and pEGFR+ processes was induced in the regrown cellular population, leading to a switch from EGFR− to EGFR+ state. Although in this study we show how switches from one state to another can be prevented, their possible advantage in the development of treatment strategy might be explored. For example, it would be interesting to examine whether incomplete primary treatments may generate less stable states that can be targeted efficiently using secondary therapies adjusted to those states.

## 5. Conclusions

In summary, this study provides new insights into possible routes of evolution of cellular signaling states when the treatments are not tailored to the individualized network structures. We show that using drug combinations that are not adjusted accurately to the individualized signatures of TNBC, may create an evolutionary pressure on the cellular population leading to the emergence of previously undetected subpopulations. We show, for example, that this pressure may change the breast cancer phenotype from TNBC to ER+, as demonstrated in the case of MDA-MD-468 malignancy. The growth of this malignancy could be stopped only when ER-inhibitors were included in the combined therapies. However, not all TNBC show this type of plasticity. MDA-MD-231 tumor growth arrest did not initially depend on ER or EGFR inhibition. EGFR+ state developed later in response to “wrong” drug treatments. Therefore, we propose that anti-TNBC therapy should be accurately tailored to the personalized signaling signature in each tumor and that incomplete or “wrong” treatments may generate diverse evolutionary routes of TNBC tumors.

## Figures and Tables

**Figure 1 cancers-13-05009-f001:**
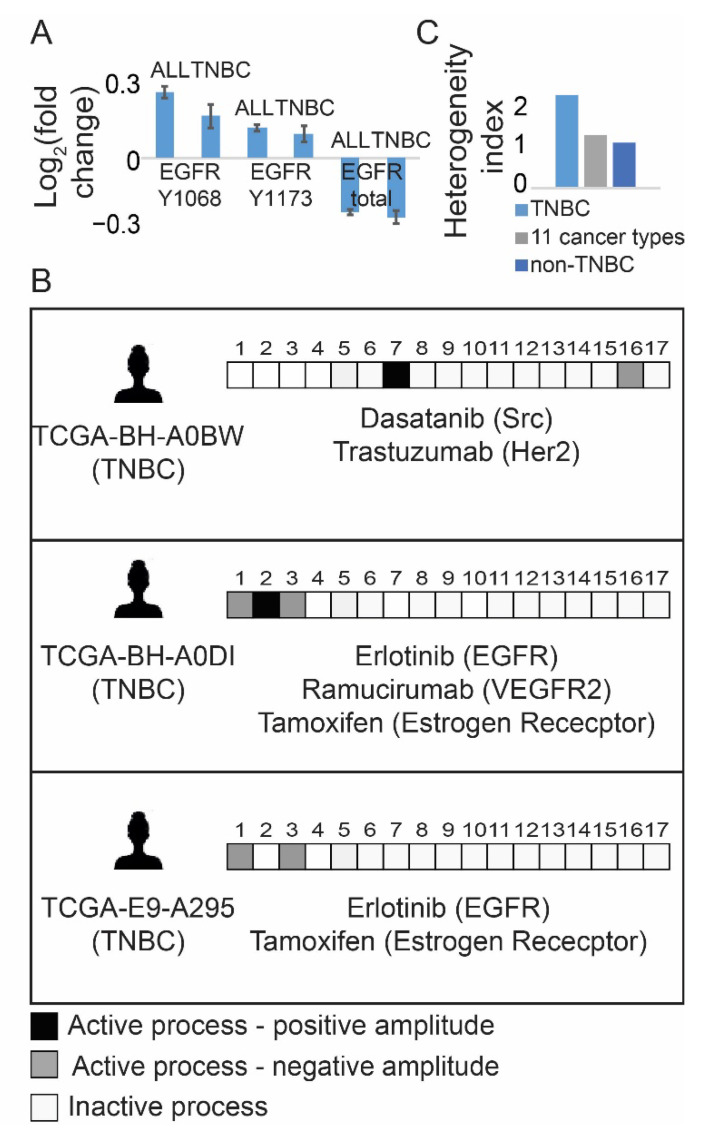
TNBC tissues are represented by different patient-specific signaling signatures, majority of which do not include EGFR. (**A**) Fold changes in expression levels of EGFR and pEGFR in TNBC and non-TNBC tumors are shown. Y1068 and Y1173 are EGFR phosphorylation sites; (**B**) Examples for patient-specific sets of active unbalanced processes are shown. Each sample harbors a set of 1–3 active unbalanced processes (PaSSS), represented schematically by a barcode. In each barcode active unbalanced processes are represented by black or gray squares, inactive white. Negative/positive amplitude denotes how the patients are correlated with respect to a particular process. Suggested PaSSS-based therapies appear below each barcode; (**C**) Heterogeneity index of 3 subgroups, represented by a ratio between the number of distinct PaSSSs and the number of samples in each subset, is shown for the TNBC subset of tissues, the entire set (3467 samples from 11 cancer types) and the subset of non-TNBC samples. (Abbreviations: TNBC—Triple Negative Breast Cancer, PaSSS—Patient-specific signaling signature, EGFR—Epidermal Growth Factor Receptor, VEGFR2—Vascular Endothelial Growth Factor Receptor 2, Her2—Human Epidermal growth factor Receptor 2, Src—Proto-oncogene tyrosine-protein kinase Src).

**Figure 2 cancers-13-05009-f002:**
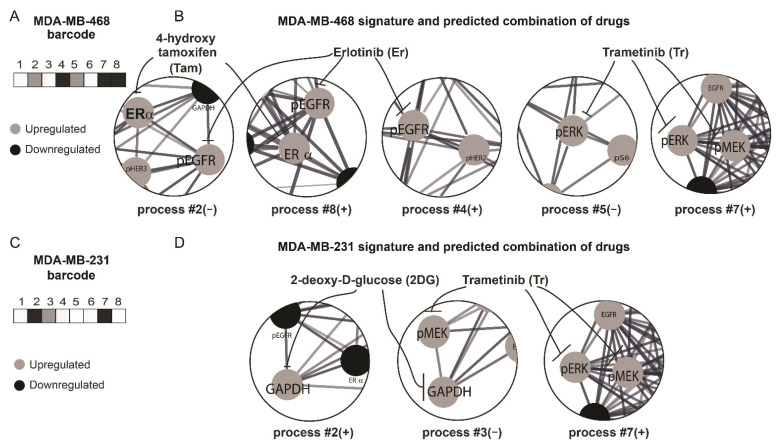
Surprisal analysis predicts efficient drug combinations for TNBC cell lines. (**A**,**B**) Barcode (**A**) representing the set of unbalanced processes in MDA-MB-468 cells, according to PaSSS analysis (**B**); (**C**,**D**) Barcode (**C**) representing the set of unbalanced processes in MDA-MB-231 cells, according to PaSSS analysis (**D**). Central protein targets from each process and the corresponding drugs (connected by inhibition arrows) are shown in B and D. (Abbreviations: TNBC—Triple Negative Breast Cancer, pEGFR—phospho Epidermal Growth Factor Receptor, PaSSS—Patient-specific signaling signature, ER—Estrogen receptor, GAPDH—Glyceraldehyde 3-phosphate dehydrogenase, pMEK—phospho Mitogen-activated protein kinase kinase, pERK—phospho Extracellular signal-regulated kinase).

**Figure 3 cancers-13-05009-f003:**
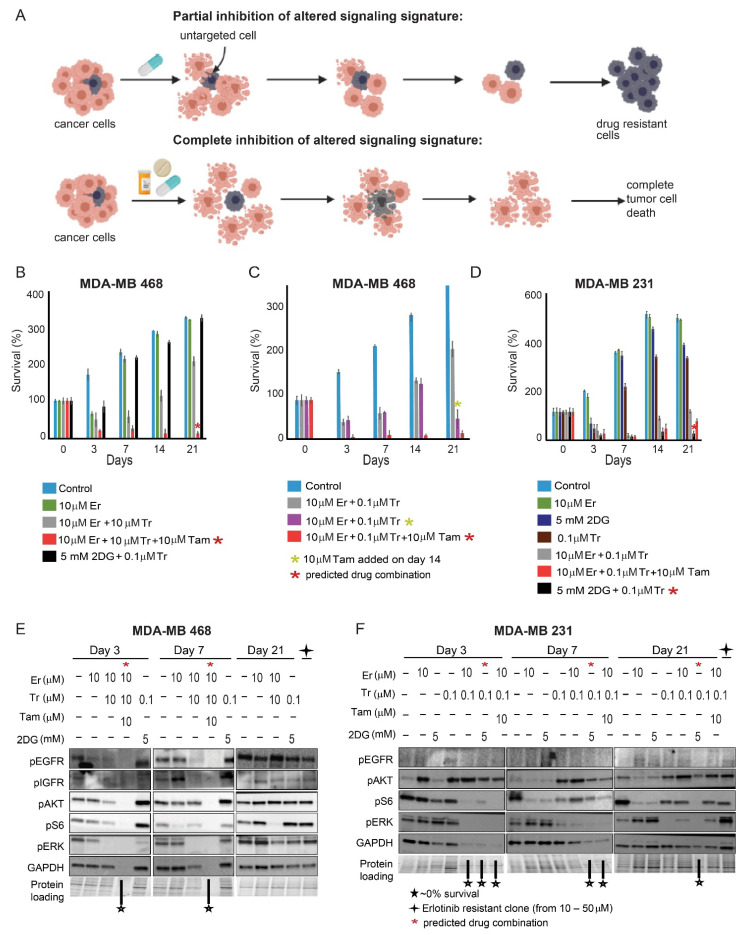
Development of resistance to various therapies. (**A**) The development of resistance to different types of therapies is shown in the illustration. Cells were treated for 72 h, allowed to regrow, and checked for survival; (**B**) In MDA-MB-468 cells, regrowth was detected when the cells were treated with Er monotherapy, Er + Tr, and also the combination predicted for MDA-MB-231 (Tr + 2DG), but not in the cells treated with the PaSSS-based therapy, Er + Tr + Tam; (**C**) Survival of MDA-MB-468 cells after the treatment with Er + Tr for 72 h. Cells were allowed to regrow until day 14, then Tam was administered on day 14 (purple columns); (**D**) MDA-MB-231 cells were treated with monotherapies, Er + Tr, Tr + 2DG, or Er + Tr + Tam, once a week for 21 days. The cells regrew after 21 days; however, no regrowth was detected when the cells were treated with Tr + 2DG; (**E**) Western blot analysis of MDA-MB-468 cells. The cells were treated with different monotherapies and combinations of drugs at different time points as indicated. Er resistant cells were developed by treating the cells with increasing amounts of Er for ~6 weeks. Due to ~0% protein content at day 21 (low survival following the PaSSS-based treatment), the results for Er + Tr + Tam treated cells are not present at day 21; (**F**) Western blot analysis of MDA-MB-231 cells. The cells were treated with different monotherapies and combinations of drugs at different time points as indicated. The Er resistant cells were developed by treating the cells with increasing amounts of Er for ~6 weeks. The results presented here represent at least 3 independent experiments. (Abbreviations: Er—Erlotinib, Tr—Trametinib, Tam—Tamoxifen, 2DG—2 deoxyglucose, pEGFR—phospho Epidermal Growth Factor Receptor, pIGFR—phospho Insulin Growth Factor Receptor, pAKT—phospho Protein kinase B, pS6—phospho Ribosomal protein S6, GAPDH—Glyceraldehyde 3-phosphate dehydrogenase, pERK—phospho Extracellular signal-regulated kinase, PaSSS—Patient-specific signaling signature).

**Figure 4 cancers-13-05009-f004:**
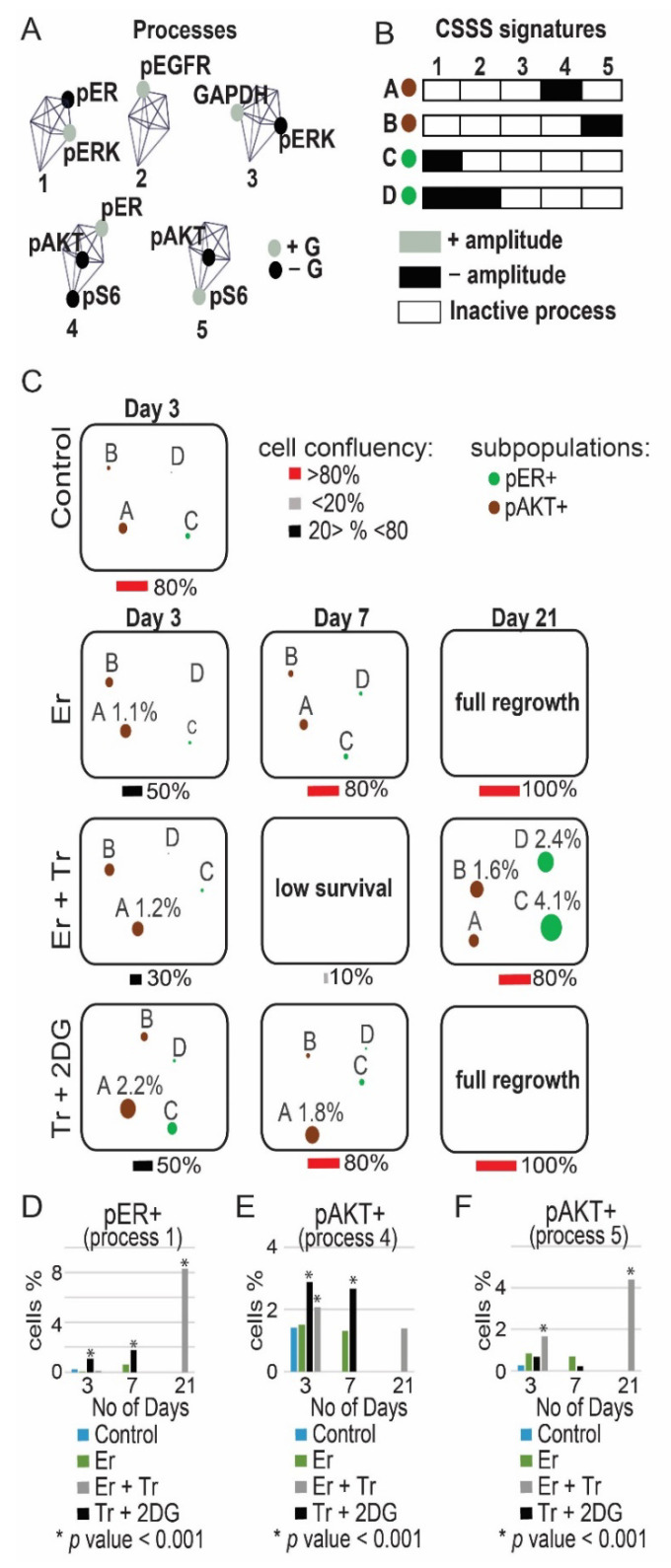
Map of the evolutionary trajectory of MDA-MB-468 cells revealed a switch in signaling state following partial treatments. MDA-MB-468 cells were treated with Er monotherapy/combination of drugs (Er + Tr, Tr + 2DG) for 72 h and allowed to regrow. Then CSSS analysis of single-cell flow cytometry data was performed to map subpopulations that expand in response to different treatments. Cell-specific signaling signature (CSSS) was assigned to each cell which divided the tumor mass into distinct cellular subpopulations. Subpopulations of cells with pAKT+ and pER+ processes, comprising more than 1% in any of the experimental conditions, are presented and quantified. (**A**) 5 unbalanced processes, as identified by the analysis, with the central proteins (those having most significant G values, representing the extent of the participation of a protein in each process) are shown. Proteins colored black are anti-correlated with those colored grey. (**B**) Cells with the same set of unbalanced processes (CSSS) are grouped into subpopulations as represented by different barcodes. Active processes are labeled by either black or grey colors. Each barcode (subpopulation) is then color-coded and presented in (**C**). Proteins labeled in black in (**A**) are induced in black processes (**B**). Proteins labeled in grey in (**A**) are induced in grey processes (**B**); (**C**) The evolution of each subpopulation was then followed over time. The size of the circle represents the percentage of each subpopulation; (**D**–**F**) Overall percentage of pER+ and pAKT+ cells was quantified for each treatment. Quantification of the subpopulations was performed using at least ~30,000 cells from each condition, which were obtained from at least 3 wells and from at least 2 independent experiments for each time point. *p*-values for each treatment vs. control are presented. (Abbreviations: Er—Erlotinib, Tr—Trametinib, 2DG—2 deoxyglucose, pEGFR—phospho Epidermal Growth Factor Receptor, pER—phospho Estrogen Receptor, pAKT—phospho Protein kinase B, pS6—phospho Ribosomal protein S6, GAPDH—Glyceraldehyde 3-phosphate dehydrogenase, pERK—phospho Extracellular signal-regulated kinase, CSSS—Cell-specific signaling signature).

**Figure 5 cancers-13-05009-f005:**
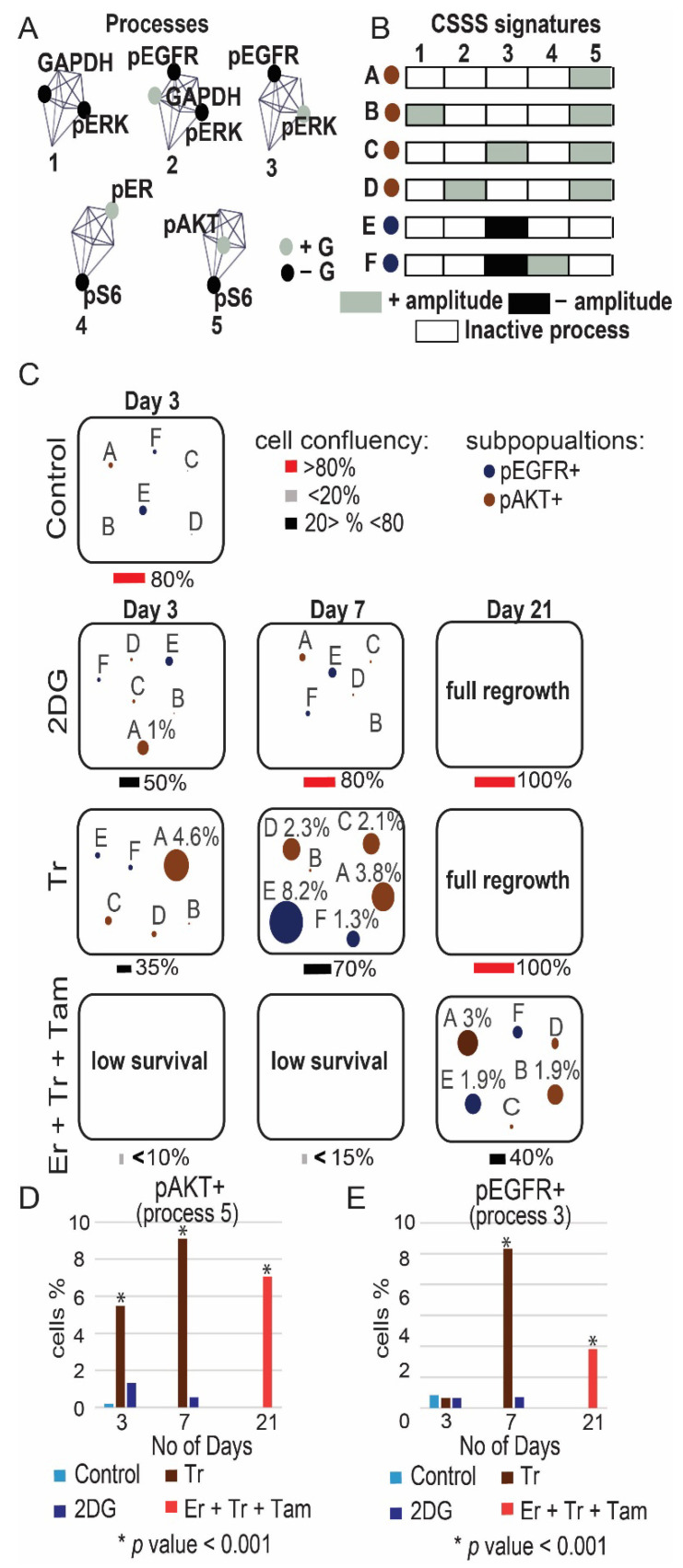
Map of the evolutionary trajectory of MDA-MB-231 cells revealed a switch in signaling state following partial treatment. MDA-MB-231 cells were treated with monotherapies/combination of drugs (Tr, 2DG and Er + Tr + Tam) once a week for 21 days and allowed to regrow. Then, CSSS analysis of single-cell flow cytometry data was performed to map subpopulations that expand in response to different treatments. Cell-specific signaling signature (CSSS) was assigned to each cell which divided the tumor mass into distinct cellular subpopulations. Subpopulations of cells with pAKT+ and pEGFR+ processes comprising more than 1% in any of the experimental conditions are presented and quantified. (**A**) 5 unbalanced processes, as identified by the analysis, with the central proteins (those having most significant G values, representing the extent of the participation of a protein in processes) are shown. Proteins colored black are anti-correlated with those colored grey. (**B**) Cells with the same set of unbalanced processes are grouped into subpopulations as represented by different barcodes. Active processes are labeled by either black or grey color. Each barcode (subpopulation) is then color-coded and presented in (**C**). Proteins labeled in black in (**A**) are induced in black processes (**B**). Proteins labeled in grey in (**A**) are induced in grey processes (**B**). (**C**) The evolution of each subpopulation was then followed over time. The size of the circle represents the percentage of subpopulation of cells. (**D**–**F**) Percentage of pEGFR+ and pAKT+ cells was quantified for each treatment. Quantification of the subpopulations was performed using at least ~30,000 cells from each condition, which were obtained from at least 3 wells and from at least 2 independent experiments for each time point. *p*-values for each treatment vs. control are presented. (Abbreviations: Er—Erlotinib, Tr—Trametinib, 2DG—2 deoxyglucose, Tam—Tamoxifen, pEGFR—phospho Epidermal Growth Factor Receptor, pER—phospho Estrogen Receptor, pAKT—phospho Protein kinase B, pS6—phospho Ribosomal protein S6, GAPDH—Glyceraldehyde 3-phosphate dehydrogenase, pERK—phospho Extracellular signal-regulated kinase, CSSS—Cell-specific signaling signature).

**Figure 6 cancers-13-05009-f006:**
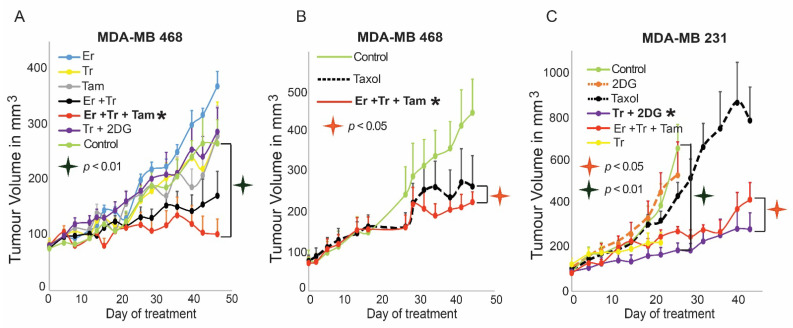
The PaSSS-based drug combinations stopped tumor growth in vivo. MDA-MB-468 (**A**,**B**) or MDA-MB-231; (**C**) cells were injected subcutaneously into mice, and once tumors reached 80 mm^3^, treatments were initiated. In both cases, the PaSSS-based drug combinations, predicted to collapse the cell line-specific altered signaling signature, inhibited the tumor growth and demonstrated an effect superior to monotherapies or combinations predicted to target partially the PaSSS (see Figure 2 for details regarding the altered signaling signatures and the PaSSS-based drug combinations). The values are mean ±  S.E. for at least *n* = 8 for each condition. (Abbreviations: Er—Erlotinib, Tr—Trametinib, 2DG—2 deoxyglucose, Tam—Tamoxifen, Taxol—Paclitaxel, PaSSS—Patient- specific signaling signature, *—Predicted combination, S.E. —Standard error, *n*—no of mice).

## Data Availability

The human tumor datasets that support the findings of this study are publicly available for download from the TCPA portal [37], https://tcpaportal.org/tcpa/download.html (accessed on 1 October 2019). Cell line dataset is available in [18].

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
