# Peer review of "Drug-Induced Resistance and Phenotypic Switch in Triple-Negative Breast Cancer Can Be Controlled via Resolution and Targeting of Individualized Signaling Signatures"

_cancers, 2021, doi:10.3390/cancers13195009_

Round 1

Reviewer 1 Report

Thanks for the attention to the issues I raised in my previous review. All my concerns have been more than addressed. 

Reviewer 2 Report

I have no further comments

This manuscript is a resubmission of an earlier submission. The following is a list of the peer review reports and author responses from that submission.

Round 1

Reviewer 1 Report

I thank the authors for the opportunity to review their work. I found their  approach to identifying unregulated networks very interesting and greatly appreciated the follow-up studies in cell lines and mice to support the bioinformatic/algorithmic results. This is an important contribution to the field of personalized cancer therapy and I look forward to hearing about its use in the clinic.

I have some suggestions for the manuscript, which I hope are easily addressable,  that I think would greatly strengthen its impact.  These concerns largely lie within issues of repeatability and figure interpretability. I separately outline them below:

Repeatability/Reproducibility Concerns
How many times was the alternative subpopulation resistance growth analysis in section 3.4 repeated? If replicated, confidence intervals (or some measure of the experiment variability) of the percentages  should be included in Table S2, if not, the experiment should be replicated at least twice more. While the use of purely technical replicates (i.e. multiple wells of cells processed in parallel one time) happens frequently, and can address some potential variance issues, we find it is better to repeat the entire experiment on different days to capture real world variance and best show a replicable result.

What informed the choice of the two out of ten cell lines in section 3.3.? The paper would be greatly strengthened by the inclusion of more of the other lines. Similarly, these cell line experiments should be repeated to confirm the results.

Figure Interpretability
Figure 2
In addition to red/blue being undistinguishable to many color blind people, the lines indicating repression and gene names on Figure 2 are hard to read. Please adjust colors to make this figure easier to better distinguish figure components and ease interpretation

Figures 4 and 5 are hard to interpret, particularly part C. For example, the inclusion of the control result on the left makes it difficult to tell that 3 cell lines to its right should be followed through time, left to right. Please modify to have a consistent flow and placing of plots throughout. An easy way to do so might  be to place the Control results under day 3 under the cell lines. The cell confluency legends in C  (bar plot and >80% and <20% diagram) should also be combined somehow as they are currently semi-redundant and confusing. The identity of A ->K in the illustrations in C should also be explained. 

Labeling the Figure 4  A->K plots with percentages or mentioning Table S2 earlier in the text (in the paragraph starting at line 346 on page 10 for example)  might also help with interpretation.  

Minor concerns
2-43 For native reader, is TNBC a basal-like breast cancer?
5-28 Would be helpful to more directly state central proteins were selected

Reviewer 2 Report

Vasudevan and coworkers present their data on targeted therapy of TNBC. The authors showed that targeting of multiple different signaling pathways is very effective based on individual patient-specific signaling signatures of the respective tumor.

The manuscript is written well and the data are very impressive. I wonder, if the authors may be willing to provide a small flowchart for the use of these data, so that an attending physician  may be able to apply some of these findings in individual therapy.
